# The Gut Mycobiome and Nutritional Status in Paediatric Phenylketonuria: A Cross-Sectional Pilot Study

**DOI:** 10.3390/nu17152405

**Published:** 2025-07-23

**Authors:** Malgorzata Ostrowska, Elwira Komoń-Janczara, Bozena Mikoluc, Katarzyna Iłowiecka, Justyna Jarczak, Justyna Zagórska, Paulina Zambrzycka, Silvia Turroni, Hubert Szczerba

**Affiliations:** 1Department of Biotechnology, Microbiology and Human Nutrition, University of Life Sciences in Lublin, 20-704 Lublin, Poland; malgorzata.ostrowska@up.lublin.pl (M.O.); elwira.komon.janczara@up.lublin.pl (E.K.-J.); 2Department of Pediatrics, Rheumatology, Immunology and Metabolic Bone Diseases, Medical University of Bialystok, 15-274 Bialystok, Poland; bozenam@mp.pl; 3Nutrition Clinic, Department of Clinical Dietetics, Medical University of Lublin, 20-093 Lublin, Poland; katarzyna.ilowiecka@umlub.pl; 4Laboratory of Regenerative Medicine, Preclinical Research and Technology Center, Medical University of Warsaw, 02-097 Warsaw, Poland; justyna.jarczak@wum.edu.pl; 5Department of Food and Nutrition, Medical University of Lublin, 20-093 Lublin, Poland; justyna.zagorska@umlub.pl; 6Metabolic Clinic, Department of Pediatrics, Rheumatology, Immunology and Metabolic Bone Diseases, Ludwig Zamenhoff University Children’s Clinical Hospital, Medical University of Bialystok, 15-274 Bialystok, Poland; paulina.zambrzycka@udsk.pl; 7Unit of Microbiome Science and Biotechnology, Department of Pharmacy and Biotechnology, University of Bologna, 40126 Bologna, Italy; silvia.turroni@unibo.it

**Keywords:** phenylketonuria (PKU), gut mycobiome, internal transcribed spacer (ITS) sequencing, phenylketonuria diet, paediatric nutrition, fungal dysbiosis

## Abstract

Background: Phenylketonuria (PKU) is a metabolic disorder managed through a strict, lifelong low-phenylalanine diet, which may influence gut microbiome dynamics. While gut bacterial alterations in PKU are increasingly investigated, the fungal community (mycobiome) remains largely unexplored. This study compared gut mycobiome composition and dietary profiles of paediatric PKU patients and healthy controls, stratified by age (<10 and 10–18 years). Methods: Stool samples from 20 children (10 PKU, 10 controls) were analysed using ITS1/ITS2 amplicon sequencing. Nutritional status was assessed using Body Mass Index percentiles (Polish standards), and nutrient intake was evaluated from three-day dietary records compared to national reference values. Correlations between fungal taxa and dietary factors were explored. Results: Although alpha diversity did not differ significantly, beta diversity and LEfSe analyses revealed distinct fungal profiles between PKU patients and controls, indicating a trend toward group separation (PERMANOVA: F = 1.54646, *p* = 0.09; ANOVA: *p* = 0.0609). PKU patients showed increased Eurotiales (*p* = 0.029), *Aspergillaceae* (*p* = 0.029), and *Penicillium* (*p* = 0.11) and decreased *Physalacriaceae* (0% vs. 5.84% in controls) and *Malassezia* (*p* = 0.13). Spearman’s analysis showed significant correlations between *Geotrichum* and intake of protein (*ρ* = 0.55, *p* = 0.0127) and phenylalanine (*ρ* = 0.70, *p* = 0.0005). Conclusions: Dietary treatment in PKU is associated with age-dependent shifts in the gut mycobiome, notably increasing the abundance of taxa such as Eurotiales, *Aspergillaceae*, and *Penicillium*, involved in carbohydrate/lipid metabolism and mucosal inflammation. These findings highlight the potential of gut fungi as nutritional and clinical biomarkers in PKU.

## 1. Introduction

In recent years, increasing attention has been given to the gut microbiota, leading to the discovery of the significant role of the fungal component, known as the mycobiome, in maintaining metabolic homeostasis and modulating the host immune response [1,2]. Although the mycobiome constitutes only a minor fraction of the gut microbial ecosystem in healthy individuals, accumulating evidence indicates its substantial contribution to the pathophysiology of various disorders, including type 2 diabetes mellitus (T2DM) and non-alcoholic fatty liver disease (NAFLD) [3,4]. Unlike the extensively investigated bacterial component of the gut microbiota, the functional relevance of gut-resident fungi is poorly characterised, particularly in the context of rare, genetically inherited metabolic conditions such as phenylketonuria (PKU).

PKU (OMIM 261600) is a rare autosomal recessive metabolic disease caused by mutations in the gene encoding phenylalanine hydroxylase (PAH), an enzyme responsible for catalysing the hydroxylation of phenylalanine (Phe) to tyrosine (Tyr) in the liver. A deficiency or absence of PAH activity leads to systemic accumulation of Phe in both the bloodstream and the central nervous system, which can result in progressive neurotoxicity if left untreated. According to global estimates, approximately 0.45 million individuals were affected by PKU in 2020, corresponding to a mean birth prevalence of 1:23,930 live births. Prevalence rates vary significantly between populations, ranging from 1:4500 in Italy to 1:8309 in Poland, 1:15,924 in China, and as low as 1:125,000 in Japan. This global variability underscores the relevance of investigating PKU from an international perspective, particularly when considering the shared dietary strategies used worldwide. Screening for PKU in Poland was first implemented in 1964 in the Warsaw region and central Poland. Since 1986, measurement of Phe levels in dried blood spots has been mandatory as part of the nationwide newborn screening program. Since May 1994, universal newborn screening for PKU has been implemented as part of the Polish national metabolic screening program, enabling early identification and timely initiation of dietary management [5]. The cornerstone of PKU treatment remains a lifelong low-phenylalanine diet, initiated early in infancy. This dietary intervention, supported by phenylalanine-free amino acid formulas and low-protein foods, is essential for maintaining metabolic control and preventing neurotoxicity. Although adjunct therapies such as large neutral amino acids (LNAA) have been explored, dietary management continues to be the gold standard of care, ensuring normal neurodevelopment and life expectancy comparable to the general population [6]. Timely initiation of dietary management in individuals with PKU is essential to mitigate the risk of irreversible neurological sequelae. Therapeutic dietary intervention is typically implemented immediately after diagnosis in the neonatal period and is critical for the prevention of cognitive dysfunction. The dietary regimen must be individually adjusted based on the patient’s Phe tolerance, which reflects their residual enzymatic activity. Maintaining plasma Phe concentrations within target therapeutic ranges requires regular biochemical monitoring and is paramount for preventing neurotoxicity. In the absence of early and sustained intervention, patients commonly exhibit intellectual disability, seizure activity, and wide range of neuropsychiatric symptoms [7].

Emerging evidence suggests that disturbances in tryptophan metabolism, alongside dysregulation of gut microbiota, may contribute to a wide spectrum of disorders associated with the gut–brain axis (GBA) [8]. In the context of PKU, it has been suggested that altered tryptophan metabolism—potentially microbiota-dependent—may affect patients’ cognitive and behavioural functions [8]. As many yeasts ferment tryptophan and may affect its bioavailability, it cannot be excluded that gut fungi are involved in this process. However, to date, there is a lack of studies evaluating the composition, function and metabolic activity of the mycobiome in PKU patients. A growing body of literature has focused on defining the optimal composition of the intestinal microbiota and characterising the metabolic outputs of microbial communities under physiological conditions [9]. In parallel, there is increasing scientific interest in the association between fungal community structure and the pathogenesis of neuropsychiatric disorders, particularly those of neurodegenerative origin [10]. Alterations in the composition of gut fungal communities observed in type 2 diabetes mellitus, obesity, and non-alcoholic fatty liver disease suggest a potential contributory role of the mycobiome in the pathogenesis of these metabolic disorders [3,4]. However, in the context of phenylketonuria, reports are still limited. Recent studies have highlighted the dualistic roles of fungal genera such as *Candida* and *Saccharomyces*, whose metabolic by products may either promote or disrupt host metabolic balance [11,12]. Importantly, disturbances in fungal community structure may lead to immune activation and/or accumulation of fungal-derived metabolites (i.e., candidalysin, prostaglandin E2, altenusin) by opportunistic fungal pathogens such as *Candida*, *Alternaria* or *Meyerozyma* [11]. The nutritional regimen in PKU, in conjunction with host-specific factors such as age, genetic background, and environmental exposures, is likely to influence the diversity and structure of the gut mycobiome.

While international dietary guidelines for PKU are well established, there is increasing recognition that dietary regimens may have broader systemic effects, particularly in the context of gut microbial ecosystems. Although most research has focused on bacterial microbiota, recent studies highlight the potential role of the gut mycobiome—the fungal component of the microbiota—in modulating host immunity, metabolism, and inflammatory status. However, data on the gut mycobiota in PKU populations remain extremely limited, especially in children. By examining both fungal composition and nutritional intake in a paediatric PKU cohort, our study aims to contribute new insights that may be of value to the global discussion on personalised diet–microbiome interactions in metabolic disorders.

The primary objective of this study was to characterise differences in the gut fungal composition between children with PKU and age-matched healthy controls and to assess the impact of a low-phenylalanine diet on the gut mycobiome. To avoid false positive results in the context of differences between the study groups, they were divided into age-matched subgroups within PKU and controls, according to the established guidelines [13]. This work aims not only to expand the current knowledge about the gut mycobiome of individuals with PKU but also to lay the groundwork for future therapeutic strategies, including targeted nutritional and probiotic interventions to modulate the gut mycobiome.

## 2. Materials and Methods

### 2.1. Study Population and Cohort Design

We conducted a preliminary study involving 20 children recruited from the Polish population. Participants were divided into two main study groups: 10 individuals diagnosed with PKU, and 10 individuals not associated with PKU as controls. Each group was further subdivided into two age categories: children under 10 years and those over 10 years of age (below 18). To minimise confounding variables, a pair-matching strategy was employed, matching individuals from the control group with PKU patients based on age and other relevant demographic factors such as sex and BMI. This individual-level matching was designed to facilitate the identification of differences related to health status and dietary intake [14]. Participants were enrolled through metabolic disease centres across Poland, which coordinated the recruitment of both cohorts. Inclusion criteria for the PKU group included a confirmed diagnosis via neonatal screening, age below 18 years, and availability of a recent phenylalanine (Phe) concentration measurement. For the control group, the eligibility criteria were defined as age under 18 years, confirmed absence of PKU in the neonatal period, and no association with other metabolic disorders. Exclusion criteria applied to both groups included the presence of congenital anomalies, chronic liver disease, acute or chronic gastrointestinal disorders, recent antibiotic treatment, or probiotic supplementation within the three months prior to the study. Each participant completed a detailed dietary questionnaire capturing demographic, anthropometric and clinical information (age, sex, height, weight, medical history, medication use, and lifestyle habits), as well as a three-day food diary and a food frequency questionnaire [14]. Parents or legal guardians received specific instructions on how to weigh and record meals to ensure the accuracy of dietary data. All data were fully anonymized prior to analysis by the research team.

### 2.2. Stool Sample Collection and Preservation

Stool samples were collected at the time of dietary data acquisition using the Stool DNA Collection & Stabilization Kit (Canvax, Valladolid, Spain). This all-in-one system enables fast, hygienic self-collection and contains 8 mL of a proprietary stabilisation buffer that inactivates nucleases and prevents microbial overgrowth, preserving native microbial community structure. Participants and their legal guardians received written and verbal instructions regarding hygienic self-collection procedures. The integrated sampling spoon and wide-mouth tube design ensured user-friendly, contamination-minimised collection. Samples were stored at room temperature (15–25 °C) for up to 48 h before transfer to −20 °C, where they remained until further analysis.

### 2.3. DNA Extraction, Library Preparation, and ITS Sequencing

Fungal DNA was extracted from stool samples using the HigherPurity ™ Stool DNA Isolation Kit (Canvax, Valladolid, Spain) according to the manufacturer’s protocol, and stored at −20 °C until further analysis. For DNA extraction, 100 U (10 µL) of lyticase (A&A Biotechnology, Gdańsk, Poland) was added to 1 mL of dissolved stool sample at the start of the process. After shaking, the mixture was incubated at 30 °C for 30 min. The next stages followed the manufacturer’s directions. DNA concentration was quantified using a Qubit 4.0 fluorometer (Invitrogen, Waltham, MA, USA), while purity was assessed using the NanoDrop™ 2000 spectrophotometer (Thermo Fisher Scientific, Waltham, MA, USA). Only samples with A260/A280 and A260/A230 ratios > 1.8 and DNA concentrations ≥ 5 ng/µL were included for downstream analysis. For each 25 ng DNA sample, PCR amplification of the ITS1–ITS2 region was performed using the 2× KAPA HiFi HotStart ReadyMix polymerase (Roche Kapa Biosystems, Wilmington, NC, USA) and primers containing Illumina adapter overhang sequences [15] (forward 5′ TCGTCGGCAGCGTCAGATGTGTATAAGAGACAG-[locus-specific sequence] and reverse 5′ GTCTCGTGGGCTCGGAGATGTGTATAAGAGACAG-[locus-specific sequence]) as described in the Fungal Metagenomic Sequencing Demonstrated Protocol (Illumina, San Diego, CA, USA). The PCR thermal profile was as follows: 95 °C for 3 min; 25 cycles of 95 °C for 30 s, 55 °C for 30 s, 72 °C for 30 s, and a final extension at 72 °C for 5 min. DNA amplicons were purified using Agencourt^®^ AMPure^®^ XP magnetic beads (Beckman Coulter, Brea, CA, USA) following the manufacturer’s instructions. Subsequently, the libraries were double-indexed using the Nextera XT kit according to the manufacturer (Illumina). PCR of 50 μL was performed under the following conditions: 95 °C for 3 min; 8 cycles of 95 °C for 30 s, 55 °C for 30 s, 72 °C for 30 s, and a final extension at 72 °C for 5 min. The final libraries were cleaned up using Agencourt^®^ AMPure^®^ XP magnetic beads (Beckman Coulter). Libraries were quantified using a Qubit dsDNA BR assay (Invitrogen). Library fragment size (360–390 bp) was verified using the TapeStation system (Agilent Technologies, Santa Clara, CA, USA). Libraries were pooled at equimolar concentrations, ensuring normalisation across the different samples sequenced in the same run. The final concentration was 50 pM. All libraries were sequenced with the Illumina PE 2 × 150 by iSeq 100 i1 Reagent v2 Kit (300 cycles) on the iSeq 100 Instrument (Illumina), according to the specifications of the manufacturer. As an internal control for a low-diversity library, 10% of PhiX viral DNA was added to the sample pool. Raw ITS sequencing data were deposited in the NCBI Sequence Read Archive (SRA) under accession number PRJNA1257280.

### 2.4. Bioinformatic Pipeline and Taxonomic Assignment

Bioinformatic analyses on gut mycobiome ITS (Internal Transcribed Spacer) were conducted using the QIIME 2 pipeline 2020.2 [16]. Data were demultiplexed, quality was filtered using the q2-demux plugin, and sequence denoising was carried out using DADA2 [16]. Length trimming and clustering filters were applied to generate the operational taxonomic unit (OTU) at 99% identity level and discard singletons as possible chimaeras. The taxonomic analysis was performed via the OTU classifier against the UNITE dynamic database [17] from phylum to species level.

### 2.5. Bioinformatics and Statistical Analysis of Mycobiome Composition

The samples were classified into four categories according to clinical phenotype and age: control_<10, PKU_<10, control_≥10, and PKU_≥10. Statistical analyses were performed using RStudio (version 2024.12.1.563) (http://www.rstudio.com/) with R version 4.4.2 (developed by the Foundation for Statistical Computing, Vienna, Austria, and utilised by RStudio, Inc., Boston, MA, USA) [18] to identify differences in clinical, demographic, anthropometric and nutritional data between matched samples. The obtained OTU table, taxonomy table, and metadata were imported into phyloseq (version 1.50.0) [19]. Mean relative abundances for each taxonomic group were visualised using horizontal bar charts with ggplot2 (version 3.5.2) [20]. A Wilcoxon rank-sum test was used to compare fungal taxon abundances between groups. All *p*-values <0.05 were considered significant. A heatmap illustrating the distribution of the most prevalent genera in individual patients was generated with the pheatmap software (version 1.0.12) [21]. Hierarchical clustering was utilised for both taxa and samples employing Euclidean distance and full linkage.

Functions from the MicrobiotaProcess package (version 1.19.0) [22] and vegan (version 2.6-10) [23] were utilised to perform diversity analyses, including alpha and beta diversity assessments, PERMANOVA, and differential abundance tests following data preprocessing. We calculated six standard diversity indices: Observed Species Richness, Chao1, Abundance-based Coverage Estimator (ACE), Shannon, Simpson and Pielou evenness. Diversity at the sample level was visualised using violin plots overlaid on boxplots, implemented using ggplot2 (version 3.5.2) [20] and ggpubr (version 0.6.0) [24]. To investigate beta diversity and visualise the overall microbial community structure, a principal component analysis (PCA) was performed using the MicrobiotaProcess package (version 1.19.0). This analysis facilitated the identification of clustering patterns among the sample groups. The input abundance table was normalised using the Hellinger transformation prior to ordination.

Differential taxonomic analysis across the four subgroups (control_<10, PKU_<10, control_≥10, PKU_≥10) was performed using the Kruskal–Wallis test. When significant differences were detected, pairwise Wilcoxon rank-sum post hoc tests were conducted to identify specific group-level differences. A circular cladogram was generated to visualise significantly different taxa in the phylogenetic tree. A taxonomic comparison was performed using the linear discriminant effect size analysis (LEfSe) method [22,25]. The analysis identified taxa that significantly differed in abundance between treatment groups (PKU vs. control) and estimated the effect size of each discriminant trait using linear discriminant analysis (LDA). Statistical tests were performed using non-parametric Wilcoxon rank sum tests, and the LDA score was estimated using a significance threshold of *p* < 0.05, and a false discovery rate (FDR) correction was used to adjust for multiple comparisons using the Benjamini–Hochberg procedure.

To identify taxa that are prevalent and abundant across several groups, the frequency (the proportion of samples in which a specific genus was present) and the mean relative abundance were computed using the microbiome package (version 1.28.0) [26]. The metrics were converted to log10, and genus-level scatter plots were produced using ggplot2. Distinct frequency–abundance plots were generated for the groups control_<10, control_≥10, PKU_<10, and PKU_≥10 to highlight specific fungal signatures linked to treatment. The analysis of microbiological patterns was conducted using the Pattern Search module in MicrobiomeAnalyst2.0 software (https://www.microbiomeanalyst.ca/, accessed on 5 November 2024) [27]. A 15% prevalence filter was applied, retaining only those taxa that were present in at least 15% of the samples. For correlation analysis, Spearman’s rank coefficient was used to evaluate associations between the relative abundance of fungal genera and selected dietary intake variables, including protein, fat, carbohydrates, fibre, saturated fatty acids (SFA), and phenylalanine. Correlations with an absolute Spearman coefficient (*ρ*) greater than 0.3 and a *p*-value below 0.05 were considered potentially meaningful. To account for multiple comparisons, FDR adjustment was applied using the Benjamini–Hochberg procedure.

### 2.6. Assessment of Nutritional Status and Dietary Intake

Nutritional status was assessed based on Body Mass Index (BMI) values, which were compared with current Polish reference standards, expressed as percentile grids developed for children of both sexes aged 1.4–17.5 years [28].

The nutritional content of consumed food items and meals was analysed using Aliant software (Cambridge Diagnostics, Poland), a commercially available program, which incorporates with the official Polish food composition database (version 6.0, software version 61) [29]. Dietary records were completed by parents or legal guardians of individuals with PKU and entered into the program by a certified dietitian (KI).

Three-day dietary records collected from the parents/guardians of individuals with PKU were used to calculate daily energy intake and nutrient consumption. These values were compared with dietary reference intakes for the Polish population, appropriate to the participants’ age groups. Additionally, nutrient intake was assessed relative to the Estimated Average Requirement (EAR), where applicable [30]. In the absence of EAR values, adequate intake (AI) was used as the reference.

A deviation of ±10% from the reference value was considered acceptable. Nutrient intake between 90% and 110% of the EAR was classified as adequate, intake below 90% was insufficient, and intake above 110% was excessive. When AI was used as the reference, intake at or above the AI was interpreted as sufficient, while intake below the AI was considered inadequate.

### 2.7. Statistical Analysis of Nutritional Data

The Shapiro–Wilk test was used to assess the normality of the distribution for continuous variables. Depending on the distribution, comparisons between two means were conducted using either Student’s *t*-test (for normally distributed data) or the non-parametric Mann–Whitney U test (for data that did not meet the assumption of normality).

## 3. Results

### 3.1. Nutritional Assessment

The study involved 20 children (10 in the PKU group and 10 in the control group). At the beginning of the study, the mean BMI of the participants was 18.5 ± 3.5 kg/m^2^ in the PKU group and 18.9 ± 4.2 kg/m^2^ in the control group. No statistically significant differences were observed in anthropometric parameters, age and gender distribution between the two groups (Table 1). The nutritional status of the participants was assessed based on the Polish reference values established by the OLA/OLAF programs, applicable to individuals aged 18 years or younger [28]. Among the respondents, 70.0% (n = 14) demonstrated a body weight within the normal range. The remaining 30.0% (n = 6) exhibited abnormal body weight, with 20.0% (n = 4) presenting as overweight and 10.0% (n = 2) as underweight. There were no statistically significant differences between the groups regarding gender distribution or nutritional status (Table 1).

Table 2 presents a summary of the intake of selected nutrients, as assessed from dietary records, in relation to the Estimated Energy Requirement (EER) within the control group. The analysis indicated that energy intake was within the recommended range (90–110% of the EER) for the majority of participants (70%; n = 7). The most pronounced deviation from dietary recommendations was observed in the intake of saturated fatty acids, with 90% of individuals (n = 9) exceeding the recommended levels. For most participants in the control group, the consumption of protein, fat, carbohydrates, and dietary fibre met the recommended dietary standards.

Half of the study group (n = 5) consumed excessive amounts of energy relative to their estimated requirements (Table 3). A high intake of total fat and saturated fatty acids was also observed in the majority of participants. The consumption of protein, carbohydrates, and dietary fibre in the study group was generally at a correct level.

Statistically significant differences were observed between the study and control groups in the intake of protein and phenylalanine, with the study group exhibiting notably lower mean values (*p* < 0.05). In contrast, no significant differences were observed between the groups in the intake of energy, total fat, saturated fatty acids, carbohydrates, and dietary fibre.

The supplements provided various vitamins and minerals to both the control group and the PKU group. In the control group, most participants had adequate intake of vitamins D, E, B3, folic acid, and B12, though a few showed deficiencies, especially in vitamins D and E (Appendix A). Mineral intake was mostly sufficient, with occasional deficiencies in selenium (Appendix A). In the PKU group, supplementation resulted in generally higher levels of intake. Most participants had enough or even excess amounts of vitamins and minerals. Notably, intakes of folic acid and vitamin B12 were consistently above the required levels, and minerals like zinc and iron were often consumed in high amounts, reducing the risk of deficiencies.

### 3.2. Alpha and Beta Diversity of the Gut Mycobiome in PKU and Control Groups

The children were divided into two age groups, 1.4–10 (control_<10 n = 6, PKU_<10, n = 6) and 10–18 years (control_≥10 n = 4; PKU_≥10, n = 4), to determine the composition of the gut microbiome. Some analyses were performed in two groups of subjects: control (n = 10) and PKU (n = 10). Alpha diversity indices were compared between PKU and control groups, stratified by age (<10 years and ≥10 years). Violin plots for ACE (*p* = 0.59; *p* = 0.34), Chao1 (*p* = 0.39; *p* = 0.20), Observed Species (*p* = 0.63; *p* = 0.77), Pielou’s evenness (*p* = 0.70; *p* = 0.49), Shannon (*p* = 0.39; *p* = 0.89), and Simpson (*p* = 0.39; *p* = 0.89) showed no statistically significant differences between the groups (*p* > 0.05 for all comparisons) (Figure 1A). A subsequent analysis without age stratification confirmed the absence of statistically significant differences in alpha diversity between the PKU and control groups with all indices (*p* > 0.05) (Figure 1B, Appendix A).

To explore differences in fungal community structure, principal component analysis (PCA) and principal coordinate analysis (PCoA) were conducted using Hellinger-transformed OTU data Bray–Curtis dissimilarities, respectively (Appendix A). PCA showed separation between PKU and control groups along PC1 (16.31%), mainly influenced by *Penicillium* (OTU3). PC2 (13.64%) reflected intra-group variability, with contributions from *Penicillium* (OTU3) and *Sporobolomyces* (OTU5), while PC3 (12.08%) was shaped by *Malassezia arunalokei* (OTU1, OTU11). PCoA revealed comparable trends. PCoA1 (14%) showed a negative association with *Penicillium* (OTU3) and a positive association with *Malassezia arunalokei* (OTU1). PCoA2 (13.11%) reflected variability within controls, mainly due to *Sporobolomyces* (OTU5), while PCoA3 (12.58%) indicated higher inter-individual variation in PKU, influenced by *Malassezia arunalokei* (OTU7) and *Candida* (OTU14).

Multivariate differences in fungal composition between PKU and control groups were assessed using PERMANOVA based on Bray–Curtis dissimilarity (adonis2 function, 99 permutations; Appendix A). The analysis yielded an F value of 1.54646 and a *p*-value of 0.09, indicating that treatment status (PKU vs. control) accounted for 7.91% of the total variance (R^2^ = 0.079). Although the results suggested a trend toward separation between the groups, statistical significance was not reached. To further evaluate group effects, an ANOVA was performed, revealing a moderate but non-significant difference in community composition (F = 3.9974, *p* = 0.0609). Homogeneity of multivariate dispersions (PERMDISP) was assessed via 999 permutations and similarly returned an F value of 3.9974 and a *p*-value of 0.0609, indicating just a trend towards dispersion between groups. We also applied pairwise comparisons between the PKU and the control groups. These showed an observed *p*-value of 0.0609 and a permuted *p*-value of 0.049, which suggests that the groups may have differences in fungal composition (Appendix A).

### 3.3. Gut Mycobiome Profiling in PKU and Healthy Children Across Age Groups

We compared the gut mycobiome composition between PKU and control groups stratified by age (<10 vs. ≥10 years) (Figure 2A–F; Appendix A). Basidiomycota predominated in PKU_<10, control_<10, and control_≥10 groups, whereas Ascomycota dominated in older PKU participants (PKU_≥10), though differences were not significant (Figure 2A, Appendix A). Statistically significant differences were observed in Ascomycota between PKU_<10 (21.27%) and PKU_≥10 (55.83%) (*p* = 0.019). Eurotiomycetes was markedly more abundant in PKU_≥10 (30.32%) compared to age-matched controls (0.58%) and PKU_<10 (4.24%) (*p* < 0.05) (Figure 2B, Appendix A). In contrast, the abundance of Malasseziomycetes tended to be higher in control_<10 (41.05%) compared to PKU_<10 (19.16%) (*p* = 0.130). In the control_≥10, the most common class was Dothideomycetes (10.05% at PKU_≥10 vs. 20.29% at control_≥10, *p* = 0.880) (Figure 2B, Appendix A). Malasseziales was most abundant in control_<10 (41.05%), with a significant decline in control_≥10 (11.32%, *p* = 0.013), though differences between control_<10 and PKU_<10 (19.16%) were not significant (*p* = 0.130) (Figure 2C, Appendix A). In PKU_≥10, Eurotiales was significantly increased (30.13% vs. 0.58% in control_≥10, *p* = 0.029), while Dothideales was more abundant in control_≥10 (15.92% vs. 0%, *p* = 0.067) (Figure 2C, Appendix A). *Malasseziaceae* predominated in children <10, especially in controls, though not significantly (*p* = 0.130) (Figure 2D, Appendix A). *Aspergillaceae* was significantly enriched in PKU_≥10 (30.13%) compared to control_≥10 (0.58%, *p* = 0.029) and PKU_<10 (*p* = 0.038) (Figure 2D, Appendix A). The group control_≥10 was characterised by an increase in the relative abundance of *Debaryomycetaceae* (0.08% at PKU_≥10 vs. 1.11% at control_≥10, *p* = 0.026) and *Pleosporaceae* (0.18% at control_<10 vs. 4.05% at control_≥10, *p* = 0.013), compared to PKU. On the other hand, in genus were higher amounts of *Candida* (6.18% at PKU_<10 vs. 0.10% at control_<10, *p* = 0.230) and *Penicillium* (28.10% at PKU_≥10 vs. 0.10% at control_≥10, *p* = 0.110) in the PKU group, but this difference was not statistically significant (Figure 2E, Appendix A). *Malassezia* was the most abundant genus in both groups with a predominance in controls (19.16% at PKU_<10 vs. 41.05% at control_<10, *p* = 0.130). We observed *Physalacriaceae_gen_Incertae_sedis* in the control_<10 group (5.68%), but not in PKU_<10. *Malassezia arunalokei* was the dominant species in both age groups, with a notable decline in relative abundance among older control children (*p* = 0.013). Also, control_<10 children were characterised by a significant increase in the relative abundance of *Physalacriaceae_*sp. (5.68%). In contrast, *Alternaria alternata* (4.57% at PKU_≥10 vs. 3.33% at control_≥10) was the second dominant species in groups above age 10 (Figure 2F, Appendix A). These findings suggest age- and condition-specific alterations in fungal composition, particularly a marked enrichment of Eurotiomycetes and *Aspergillaceae* in older PKU patients, potentially reflecting long-term dietary effects.

To complement the observed age- and condition-specific shifts, a heatmap of the 25 most prevalent fungal taxa was built (Figure 3). Several taxa were notably more prevalent among PKU and control patients, including Helotiales (PKU1), *Penicillium* (PKU1, PKU6, C6), *Candida* (PKU2, PKU7), Dothideales (C1), *Sporobolomyces* (PKU5, C2, C7) (*Physalacriaceae_*sp (C2, C4, C7, C9), and *Malassezia arunalokei* (PKU1, PKU3, PKU8, C2, C3, C5, C7, C8) (Figure 3).

### 3.4. Core Gut Mycobiota Structure and Age-Dependent Shifts in Children with Phenylketonuria and Healthy Controls

Across all study groups, *Malassezia* species, particularly *Malassezia arunalokei* and *Malassezia globosa,* were consistently among the most prevalent and abundant fungal taxa (Figure 4A–D). In the control group, *Sporobolomyces* (both_<10 and_≥10 years), *Aspergillaceae* (control_<10) and several unclassified fungal taxa exhibited notable prevalence and abundance. In the PKU group, the dominance of *Malassezia* persisted, accompanied by a slight increase in the prevalence of *Aspergillaceae* (PKU_<10) and *Penicillium* (PKU_≥10).

### 3.5. Fungal Biomarkers Associated with PKU and Control Groups: A LEfSe Analysis

To identify fungal taxa discriminating between PKU and control groups, we employed linear discriminant analysis effect size (LEfSe) (Appendix A). The logarithmic LDA score was used to estimate the effect size, with higher scores indicating stronger discriminatory power of the taxon. The data in Appendix A are presented in the form of a cladogram (Figure 5A) and an LDA score plot (Figure 5B). Several taxa were significantly enriched in the control group, including *Physalacriaceae* (mean LDA: 4.29, *p* = 0.0052), *Geotrichum candidum* (mean LDA: 3.33, *p* = 0.0019), *Agaricaceae*, *Agaricus* (mean LDA: 3.03, *p* = 0.0304), *Stereaceae*, *Stereum* (mean LDA: 2.83, *p* = 0.0304), and Basidiomycota (mean LDA: 5.01, *p* = 0.0412) (Appendix A, Figure 5A,B). Conversely, the following taxa were significantly more abundant in the PKU group: *Erysiphaceae*, Erysiphales (mean LDA: 3.59, *p* = 0.0130), *Mycosphaerellaceae* (mean LDA: 3.66, *p* = 0.0130), Eurotiomycetes (mean LDA: 4.81, *p* = 0.0210), *Aspergillaceae* (mean LDA: 4.81, *p* = 0.0233), *Blumeria, Blumeria_graminis* (mean LDA: 3.37, *p* = 0.0304), *Sawadaea, Sawadaea nankinensis* (mean LDA: 3.23, *p* = 0.0304) and Ascomycota (mean LDA: 4.97, *p* = 0.0343) (Appendix A, Figure 5A,B). While these taxa exhibited strong discriminatory potential between groups, the false discovery rate (FDR ≈ 0.29) suggests that findings should be interpreted with caution and validated in larger, independent cohorts.

### 3.6. Nutritional Correlates of Gut Mycobiome Composition

Spearman’s rank correlation analysis was employed to assess associations between specific fungal taxa and dietary variables in both PKU and control children (<10 years and ≥10 years). The results are visualised in Figure 6A–F and summarised in Appendix A. A correlation study indicated a significant association between the relative abundance of *Geotrichum* and the intake of protein (*ρ* = 0.55, *p* = 0.0127; Figure 6A, Appendix A), Phe (*ρ* = 0.70, *p* = 0.0005; Figure 6E, Appendix A), and SFA (*ρ* = 0.45, *p* = 0.0457; Figure 6F, Appendix A). In contrast, the abundance of *Entyloma* exhibited a negative correlation with the consumption of protein (*ρ* = –0.48, *p* = 0.0328; Figure 6A, Appendix A), fat (*ρ* = –0.49, *p* = 0.0284; Figure 6B; Appendix A), carbohydrates (*ρ* = –0.48, *p* = 0.0331; Figure 6C, Appendix A), and SFA (*ρ* = –0.45, *p* = 0.0484; Figure 6F, Appendix A). *Stereum* showed a positive correlation with Phe consumption (*ρ* = 0.46, *p* = 0.0408; Figure 6F, Appendix A). These relationships lost significance following FDR correction and should be regarded with caution. Other examined genera exhibited weak or non-significant correlations with nutrient intake. *Penicillium* showed no significant correlation with any dietary components (maximum *ρ* = 0.005, *p* = 0.982 for protein). *Candida* demonstrated a negative but weak correlation with protein intake (ρ = –0.26, *p* = 0.274), along with similarly weak or inconsistent associations with other macronutrients (Figure 6A; Appendix A). *Malassezia* and *Sporobolomyces* displayed weak positive correlations with various dietary variables (*ρ* < 0.22), none of which reached statistical significance (*p* > 0.36 for all comparisons). *Alternaria* showed slightly stronger, although still modest, positive correlations—especially with protein (*ρ* = 0.32, *p* = 0.176)—but could not exceed the established thresholds for significance (*ρ* > 0.3 and *p* < 0.05). In summary, these genera exhibited no statistically significant associations with dietary intake and were therefore considered biologically non-informative in this context.

## 4. Discussion

### 4.1. Nutritional Implications in PKU Management

A review of the literature highlights significant challenges related to the dietary habits of children with phenylketonuria (PKU), emphasising issues such as food selectivity, imbalances in nutrient intake, and parents stress associated with diet management.

Many studies report a high prevalence of food neophobia and selectivity among children with PKU, particularly during the introduction of new foods in early childhood [31,32]. The restrictive low-phenylalanine diet, based on a limited selection of low-protein foods and amino acid supplements, results in a narrowed repertoire of accepted meals. Children with PKU exhibit greater resistance during mealtimes and are more reluctant to try new foods compared to their healthy peers, indicating a broader issue of dietary monotony and rigid eating behaviours [31]. This conclusion aligns with our observations, in which children with PKU had less varied meal plans compared to the control group.

Regarding qualitative diet analysis, findings are mixed. While many children meet protein intake recommendations through the use of amino acid formulas, studies indicate excessive energy intake, particularly from high-carbohydrate and low-protein products. Importantly, increased risk of overweight and obesity has been noted in children with PKU—with some studies showing that up to 50% of school-aged children exceed BMI norms [33]. This may be due to compensatory consumption of energy-dense foods combined with reduced physical activity. Our findings contrast with these results—only 20% of children in the study group showed BMI values indicative of overweight. In regard to fat intake in a Latvian cohort [34], children aged 1–3 years with PKU were found to have dietary fat consumption below the national recommendations, indicating a potential need for closer nutritional monitoring in this age group. The cited results are consistent with our observations, which demonstrated that 70% of children in the study group consumed insufficient amounts of fat relative to their requirements. Protein intake—mainly from phenylalanine-free formulas—slightly exceeded the RDI (113–129% of RDI) in a U.S. study and was associated with normal linear growth, suggesting effective diet management under medical supervision [35]. Similarly, our results showed that protein intake by both the control and PKU groups was within recommended limits in almost all analysed cases. Micronutrient intake in PKU children was generally adequate, though some studies reported higher-than-recommended intakes of iron, selenium, and zinc. Notably, no adverse biochemical levels of these trace elements were observed, indicating efficient homeostatic regulation [36]. This conclusion was supported by our findings for zinc, where excessive intake was observed in 70% of the study group. Importantly, this pattern was not noted among the healthy participants. Iron intake was adequate in both groups. On the other hand, selenium intake was insufficient in the control group, whereas in children with PKU it was predominantly within the recommended range (Appendix A).

Another important issue is legal guardians’ lack of practical dietary management skills. U.S. studies have shown that despite high general knowledge about the PKU diet, legal guardians often struggle to correctly estimate phenylalanine content in foods [37]. This underlines the need for ongoing education and individualised support from professionals.

In summary, effective dietary treatment of PKU requires an interdisciplinary approach that includes not only optimisation of nutrient intake, but also psychological support, behavioural interventions, and education tailored to the needs of patients and their families. Future research should focus on longitudinal analyses to assess the long-term effects of evolving dietary patterns, especially in the context of emerging adjunct therapies and possible dietary liberalisation.

### 4.2. Comparative Multivariate Analysis of Fungal Communities in PKU and Controls

Despite evident taxonomic shifts, no significant differences in fungal alpha diversity were observed between PKU patients and healthy children, as assessed by six indices of richness, evenness, and overall diversity. This suggests that although specific taxa undergo compositional changes, the overall diversity of the gut mycobiome remains stable—highlighting the need for further functional studies. Similarly, a study of 300 children from the Bandiagara Malaria Project found no significant differences in alpha diversity among those aged 0–2 years compared to those aged 3–8 years and 9–15 years [38]. Earlier studies on an Italian population n = 111 showed that the richness of the gut fungal microbiota of infants (0–2 years) and children (3–10 years) was greater than that of adults (≥18 years). Research demonstrates that the inhibition of bacterial microbiota development during antibiotic treatment leads to an increase in the gut fungal microbiota [39]. Principal component analysis (PCA) revealed a distinct separation between the fungal communities of PKU patients and controls, indicating that the restrictive PKU diet influences gut fungal composition. In the PKU cohort, *Penicillium* appeared to drive changes in community structure, whereas the control group exhibited a more stable and homogeneous mycobiome, likely reflecting greater dietary variety. Within this group, *Malassezia arunalokei* was strongly associated with compositional variability.

Principal coordinate analysis (PCoA) confirmed these findings: *Penicillium* (OTU3) was more prevalent in PKU samples, while *Malassezia arunalokei* (OTU1) was enriched in controls. The association of OTU1 with lipid metabolism suggests a potential role in maintaining intestinal barrier integrity. Although the function of *M. arunalokei* in the gut remains largely unknown, it has been reported as a dominant commensal in the healthy ear microbiome, alongside *M. restricta*, where it may exert a protective, mutualistic effect [40]. The richness and biomass of the microbiome exhibited substantial differences between healthy and diseased ears, with studies indicating a possible mutualistic, protective role of *Malassezia species* against the disease [40]. The microbiome of the affected ear includes recognised fungal pathogens, including *Aspergillus* sp. and *Candida* sp. Additional investigation is required on the impact of PKU and dietary interventions on the decrease in *M. arunalokei*. The fact that *Candida* (OTU14) is common in people with PKU suggests that the gut ecosystem has changed, which could cause dysbiosis and inflammation [41]. *Sporobolomyces* (OTU5) induce changes in control samples, suggesting a potential association with fibre metabolism. Prior research has shown that species of *Sporobolomyces* demonstrate antibacterial characteristics. The pigment obtained from the extract of *Sporobolomyces* sp. inhibited the proliferation of *E. coli*, *S. aureus*, *S. faecalis*, *B. subtilis*, *Enterococcus* sp., and *P. aeruginosa*, exhibiting varying inhibition zones [42]. PERMANOVA analysis did not detect statistically significant differences between PKU and control groups. The low R^2^ value (0.079) suggests that other factors—such as diet, host genetics, and environmental exposures—may exert a stronger influence on fungal community composition in PKU. Prior research shows that individuals with PKU exhibit alterations in their gut microbiome [43,44,45]. The observed changes primarily result from a strict low-protein diet supplemented with specialised medical food [46]. Our findings suggest a trend toward compositional reorganisation of the gut mycobiome in PKU, warranting further investigation.

### 4.3. Fungal Community Shifts in PKU Children According to Diet and Age

A comparative analysis revealed differences between the PKU and control groups. A decrease in Basidiomycota (32.89% at PKU_<10, 17.04% at PKU_≥10) abundance was observed in the PKU compared to control (55.64% at control_<10; 33.78% at control_≥10), demonstrating a possible influence of PKU dietary restrictions on the composition of the fungal community. Similarly, a study of children with attention-deficit hyperactivity disorder (ADHD) showed a significantly higher abundance of Ascomycota and a significantly lower abundance of Basidiomycota than in a healthy control group [47], which was noticed in PKU_≥10. Since Basidiomycota species are commonly associated with food consumption and fibre degradation [48], this reduction may indicate a shift in fungal colonisation due to diet-related alterations. On the other hand, findings from other research indicate an increase in Basidiomycota and a decrease in Ascomycota, both associated with inflammatory bowel disease (IBD) activity. Differences in the proportions of phylum were noted across disease remission, disease activity, and healthy controls, suggesting that this ratio could be used as a marker for fungal dysbiosis [49]. Our study could not enable us to definitively conclude that the differences we have seen are linked to dysbiosis since 26.60% to 45.84% of the taxa remain unidentified.

We have observed a decline in Malasseziomycetes (19.16%) (*Malasseziaceae*) among PKU patients (<10 years) that suggests a reduction in dietary substrates essential for these fungi. A reduced presence could result in a decrease in fungal diversity or modifications in metabolic functions. In contrast, the PKU_≥10 group exhibited an increased level of Eurotiomycetes (30.32%). This class is poorly researched; certain dietary patterns and treatments may indirectly affect its abundance and activity. Although much attention has been directed to *Candida* (Ascomycota) [1]. It is conceivable that Eurotiomycetes may also be influenced by similar dietary patterns as members of the same phylum.

The increase in Leotiomycetes (10.99%) in PKU_≥10, known for degrading plant-derived carbohydrates, suggested an adaptation to the carbohydrate-rich PKU diet. The latest study investigated the relationship between diet, intestinal fungi, and gastrointestinal health in 46 patients with gastrointestinal disorders. The Mediterranean diet can change the fungal intestinal microbiota and improve gastrointestinal disorders, with a lower abundance of Leotiomycetes and Sordariomycetes and a significantly higher abundance of Saccharomycetes observed after the intervention [50].

In our study, we also observed an increased abundance of *Aspergillaceae* (30.13%) in PKU_≥10, which includes fungi capable of metabolising a wide range of carbohydrates. The increased prevalence of PKU patients consuming carbohydrate-rich medical foods indicates an adaptive transition towards fungi with specialised carbohydrate metabolic pathways [46]. *Penicillium* species, known for thriving in carbohydrate-rich environments, showed increased presence in PKU_≥10 patients (28.10%). The investigation, including the low-carbohydrate high-fat (LC) diet, revealed an enrichment of one genus, *Ustilaginaceae* sp., and a depletion of five genera: *Blumeria*, Agaricomycetes, *Malassezia*, *Rhizopus*, and *Penicillium* [51]. This research indicates that the LC diet may reduce the levels of *Penicillium*, among other factors. These fungi play a crucial role in carbohydrates’ fermentation and bioactive metabolite synthesis, further emphasising the impact of dietary interventions on fungal composition [52].

The *Malassezia* genus mainly functions as a coloniser and pathogen of the skin, while also being present in the intestines [53]. The majority of *Malassezia* species are unable to synthesise fatty acids and instead acquire them from external lipid sources. In the intestine, lipids are derived from bile salts synthesised from bile acids by hepatocytes [54]. Different theories exist regarding the colonisation of *Malassezia* in the human intestine, including indirect exposure to human skin or the ingestion of breast milk during infancy [54]. Recent studies indicate that *Malassezia* is the predominant taxon in fungal colonisation and plays a role in intestinal symbiosis [53]. *Malassezia* possesses pathogenic potential, capable of inducing detrimental and abnormal immunological responses, such as seborrhoeic dermatitis [55]. *Malassezia* was identified in higher amounts in patients with inflammatory bowel disease (IBD) compared to healthy controls and has been suggested to have an essential role in mucosal inflammation [56]. The overall decline in *Malassezia* (19.16% at PKU_<10) suggests a shift from lipid-processing fungi to those that specialise in carbohydrate metabolism. This transition may influence gut microbial interactions, intestinal barrier integrity, and immune responses [57,58]. The decreased prevalence of *Malassezia* species that metabolise lipids reflects the lipid-restricted nature of PKU dietary interventions. These findings underscore the importance of dietary composition in modulating fungal colonisation and host immune responses in paediatric PKU. The prevalence of *Physalacriaceae_*sp (5.68%) in healthy children (<10 year) indicates that these fungi thrive on foods that are absent or less prevalent in PKU diets. The family *Physalacriaceae*, belonging to the order Agaricales, performs a significant ecological function in lignin degradation and carbon circulation [59]. Some members of this family have been shown to break down lignin and cellulose in plant cell walls [59]. However, knowledge of *Physalacriaceae* is limited, notably in areas such as genetics and secondary metabolite synthesis, or human gut health. Our research has shown that *Candida tropicalis* (2.27%), a species associated with intestinal inflammation [60], was present in the PKU_<10 group, although this was not statistically significant. Previous studies have revealed that the diet significantly influenced the colonisation of *Candida*, with high-carbohydrate diets promoting its growth [61]. Conversely, cheese consumption may inhibit *Candida* growth through the action of saturated fatty acids [61]. Another clinical/control study conducted on 124 paediatric patients aged 2–18 identified an increased incidence of *Candida tropicalis* in people diagnosed with Crohn’s disease compared to the control group [62]. In other studies, *C. tropicalis* caused dysbiosis, resulting in modifications in the presence of mucin-degrading bacteria and changed expression of tight junction proteins, hence increasing intestinal permeability. This subsequently elicited a strong Th1/Th17 response, producing an expedited pro-inflammatory phenotype in mice with experimental colitis [63].

### 4.4. Differential Abundance of Taxa in Healthy Children and PKU

The differential abundance of specific fungal taxa suggests that phenylketonuria influences gut fungal composition, likely due to metabolic limitations and dietary restrictions. Increased occurrences of *Helotiales* (PKU_≥10), *Penicillium* (PKU_≥10), *Candida* (PKU_<10), and *Didymellaceae* (PKU_<10; PKU_≥10) in the PKU group may reflect a shift toward fungi capable of metabolising carbohydrates typical of a restricted PKU diet.

Studies have shown that the dominant mycobiome taxa in inflammatory bowel disease include *Saccharomyces*, *Penicillium*, *Aspergillaceae*, and *Candida*, including *Candida tropicalis* [45,49]. The presence of *Candida* species, which are potential opportunistic pathogens, may indicate diet-induced dysbiosis or compromised gut barrier function—patterns often observed in metabolic disorders.

The elevated abundance of *Penicillium* and *Aspergillaceae* in PKU may be associated with the intake of specialised carbohydrate-based medical foods, though further studies are required. Previous research has shown a positive correlation between *Candida* and carbohydrate consumption and a negative correlation with total saturated fatty acid intake. Similarly, *Aspergillus* negatively correlates with SFA intake [1].

Dietary intake of dairy products, a primary source of protein, was inversely associated with *Candida* and positively with *Saccharomyces* [64]. In vitro studies have demonstrated that fungal species differ markedly in their ability to metabolise fatty acids and alcohols as carbon sources [65]. Fatty acids may also inhibit fungal growth; however, data on their effect on the human gut mycobiome remain limited and inconclusive. In contrast, Dothideales (15.92% at control_≥10), *Physalacriaceae_*sp (5.68% at control_<10), and *Debaryomycetaceae* (1.11% at control_≥10) were more prevalent in healthy children. Interestingly, mycobiome profiles of bladder cancer patients undergoing chemotherapy also showed increased Dothideales compared to healthy controls [51]. Nonetheless, little is known about the dietary determinants or physiological relevance of Dothideales in humans. *Debaryomycetaceae* species are common inhabitants of the oral and gastrointestinal tract in healthy individuals, where they support digestion, modulate immune responses, and help maintain mucosal integrity [57,66]. Our data revealed substantial inter-individual variability in fungal community composition—even within the same treatment group. Some PKU patients clustered into microbiological subtypes, while others exhibited unique fungal profiles. This variation suggests that additional factors beyond diet—such as host genetics, gut physiology, and environmental exposures—play a role in shaping fungal community structure. Fungi significantly impact gut homeostasis by regulating opportunistic pathogens and contributing to nutrient fermentation. Disruptions in the mycobiome may promote dysbiosis, which has been linked to inflammatory states and chronic diseases, including IBD [41]. Further research is needed to explore dietary influences on the gut mycobiome and to map the metabolic pathways and enzymatic processes involved in fungal fermentation of dietary substrates. A deeper understanding of how specific dietary components shape fungal diversity and function is crucial to evaluating whether nutritional interventions can modulate the gut mycobiome to improve clinical outcomes in PKU.

### 4.5. LEfSe-Based Identification of Differentially Abundant Fungal Taxa in PKU and Healthy Controls

Linear discriminant analysis effect size (LEfSe) revealed significant taxonomic differences in the gut mycobiota between PKU patients and healthy controls. Taxa that were significantly enriched in the control group were primarily affiliated with the phyla Basidiomycota and Ascomycota. Within Basidiomycota, these included *Physalacriaceae*, *Agaricaceae*, *Stereaceae*, *Agaricus*, and *Stereum*. These findings are consistent with previous studies suggesting that Basidiomycota taxa, along with selected Ascomycota members, are involved in dietary fibre degradation and are typically associated with more diverse, unrestricted diets. In contrast, the PKU group exhibited enrichment of several Ascomycota taxa, including *Erysiphaceae*, *Mycosphaerellaceae*, Erysiphales, Mycosphaerellales, Eurotiomycetes, *Aspergillaceae*, Eurotiales, *Blumeria*, *Sawadaea*, *Blumeria graminis*, and *Sawadaea nankinensis*. While members of the *Erysiphaceae* family are predominantly known as phytopathogens (e.g., powdery mildews), emerging evidence suggests that some may interact with host-associated microbial communities, including in the oral cavity, potentially influencing host physiology [67]. The predominance of Ascomycota in PKU may reflect a compositional shift driven by disease-specific dietary constraints, particularly the use of low-protein, carbohydrate-rich medical foods. However, this hypothesis warrants further validation in larger cohorts. The application of FDR correction (*q* ≈ 0.29) indicates that while several taxa showed differential abundance between groups, broader confirmatory studies are needed. Wide confidence intervals observed for some taxa may reflect inter-individual variability in metabolic phenotype, dietary compliance, and gut ecosystem composition. In contrast, taxa with narrow confidence intervals exhibited more stable associations with either the control or PKU groups, suggesting a more robust biomarker potential.

### 4.6. Gut Fungal Dysbiosis in PKU: Biomarkers, Clinical Implications, and Therapeutic Perspectives

Distinct shifts in gut fungal composition underscore the potential of specific taxa as biomarkers of PKU-related dysbiosis. A notable decrease in Malasseziales, Agaricales, and Basidiomycota taxa (e.g., *Physalacriaceae, Agaricaceae, Stereaceae*) was observed in PKU patients, suggesting a depletion of taxa involved in microbial balance, fibre degradation, and gut homeostasis. Conversely, an increased prevalence of Eurotiomycetes, *Aspergillaceae, Erysiphaceae*, and *Mycosphaerellaceae* indicates significant shifts in fermentation processes, immunological responses, and carbohydrate metabolism pathways. The presence of *Geotrichum candidum* in controls further suggests its probiotic potential, possibly supporting bacterial populations and intestinal stability. The reduction in fungal diversity, especially in Basidiomycota, supports the hypothesis that dietary modifications in PKU negatively impact the gut mycobiome, leading to a loss of beneficial taxa. From a clinical standpoint, monitoring key fungal biomarkers like *Aspergillaceae, Erysiphaceae*, and *Mycosphaerellaceae* may help detect early microbiome disturbances in PKU patients. Considering the association between *Aspergillaceae* and gut inflammation, targeted probiotic or antifungal interventions may represent promising therapeutic avenues to re-establish mycobiome equilibrium. Moreover, dietary strategies aiming to promote fungal diversity may enhance gut and metabolic health. Finally, taxa such as *Penicillium*, *Candida*, *Sporobolomyces*, and *Malassezia arunalokei* emerged as additional candidates for future biomarker research. Collectively, these fungal signatures provide novel insights into PKU-associated microbial shifts and highlight promising targets for therapeutic modulation of the gut mycobiome.

### 4.7. Study Limitations

Despite the strengths of our study—such as age-matched subgroup comparisons and an integrated assessment of the gut mycobiome and dietary intake—several limitations must be acknowledged. First, the cross-sectional nature of the study precludes establishing causal relationships between dietary patterns and gut fungal composition. The relatively small sample size, particularly after stratification into age groups, may have limited statistical power and increased the risk of type II error, potentially obscuring subtle but biologically relevant differences. Moreover, dietary intake was assessed using short-term recall data, which may not accurately reflect habitual consumption and is susceptible to reporting bias, especially given its reliance on legal guardian-reported estimates of phenylalanine-free medical foods. Fungal profiling based on ITS sequencing also has intrinsic limitations, including suboptimal taxonomic resolution and the possibility of under-representation of low-abundance taxa due to the generally low fungal biomass in stool samples. In addition, although participants were drawn from a relatively homogenous clinical setting, the findings may not be generalisable to broader populations due to cultural, genetic, and environmental influences that were not accounted for in the analysis. However, potential confounding effects from recent antimicrobial or probiotic use were minimised through strict exclusion criteria: none of the participants had used antibiotics or probiotics within three months prior to sample collection. Despite this control, other unmeasured lifestyle or environmental factors may have contributed to inter-individual variation. Future research should prioritise longitudinal, multi-centre studies with broader metadata integration to further elucidate the mechanisms linking fungal dysbiosis, diet, and metabolic outcomes at PKU.

### 4.8. Future Perspectives and Methodological Considerations

The gastrointestinal tract of humans has a wide range of microorganisms that together constitute intricate and dynamic ecosystems. Several bacterial–fungal interactions associated with diseases have been reported [68]. Bacterial–fungal interactions may be either synergistic or competitive, influencing niche colonisation and nutrient acquisition and potentially contributing to microbial imbalance. Research indicates that the production of short-chain fatty acids by bacteria during fibre fermentation has antifungal effects [69]. Recent data show that during intestinal inflammation or prolonged antibiotic usage episodes, the intestinal bacterial population diminishes, leaving it susceptible to invasive fungus species [70]. Intestinal microorganisms generate an extracellular network termed biofilm to protect themselves from antimicrobial agents.

Future research should focus on advancing taxonomic classification and functional analysis through shotgun metagenomics and metatranscriptomics. Whole-genome sequencing may also be necessary for a more comprehensive taxonomic description of unknown fungal species. Understanding the functional consequences of alterations in fungal composition requires a multi-omics approach, incorporating metabolomic analyses to investigate changes in microbial metabolic pathways. This could provide valuable insights into how fungal populations contribute to PKU’s nutrient metabolism, intestinal homeostasis, and systemic health. Future studies should also focus on the interactions between fungi, bacteria, and host metabolism. These interactions may influence microbiome stability and gut function, affecting host metabolic processes. Multi-species analyses should be conducted to determine the contributions of both fungal and bacterial communities to intestinal homeostasis and immune regulation. Larger sample sizes and advanced sequencing technologies will be necessary to capture the full spectrum of microbial variation and yield more conclusive insights into the role of gut fungi in PKU. The exploration of probiotic and antifungal strategies may offer novel approaches as potential ways to control gut microbiome dysbiosis due to the fungal imbalances seen in PKU. The targeted modulation of fungal populations may help restore microbiome balance and reduce inflammation or metabolic dysfunction associated with PKU. Investigating the relationship between fungi, bacteria, and host metabolism will provide critical insights into microbiome-driven mechanisms, potentially leading to new therapeutic and dietary strategies for improving gut health in individuals with PKU.

We understand that comprehensive and well-maintained reference databases are essential for mycobiome research. Our study found unclassified fungal sequences in both the PKU and control groups. This shows that there are not enough databases for classifying fungal taxonomy. Our research identified many fungi at different taxonomic levels using the UNITE database. It is important to acknowledge that the fungal nomenclature system is now undergoing reorganisation, and there are many fungi still to be identified and characterised [17].

## 5. Conclusions

This pilot investigation offers new insights into the gut mycobiome composition of children with phenylketonuria (PKU), indicating that long-term dietary control correlates with specific, age-dependent alterations in fungal communities. Notwithstanding similar alpha diversity, the PKU group exhibited a heightened prevalence of carbohydrate-adapted taxa, including *Aspergillaceae* and *Penicillium*, alongside a diminished presence of lipid-associated taxa such as *Malassezia* spp., indicative of the effects of a phenylalanine-restricted, carbohydrate-rich diet. Our findings correspond with increasing data in the literature indicating that dietary interventions in metabolic diseases may substantially alter the gut fungus ecosystem. Comparable compositional changes have been documented in other populations with modified diets or chronic illnesses, highlighting a possible global trend in diet–mycobiome interactions. This study offers a distinctive viewpoint by delineating these associations explicitly in paediatric PKU, a rare condition with stringent dietary constraints. Future studies should aim to functionally validate these correlations, explore their longitudinal stability, and evaluate the clinical potential of mycobiome-targeted interventions to improve metabolic and gastrointestinal outcomes in PKU.

## Figures and Tables

**Figure 1 nutrients-17-02405-f001:**
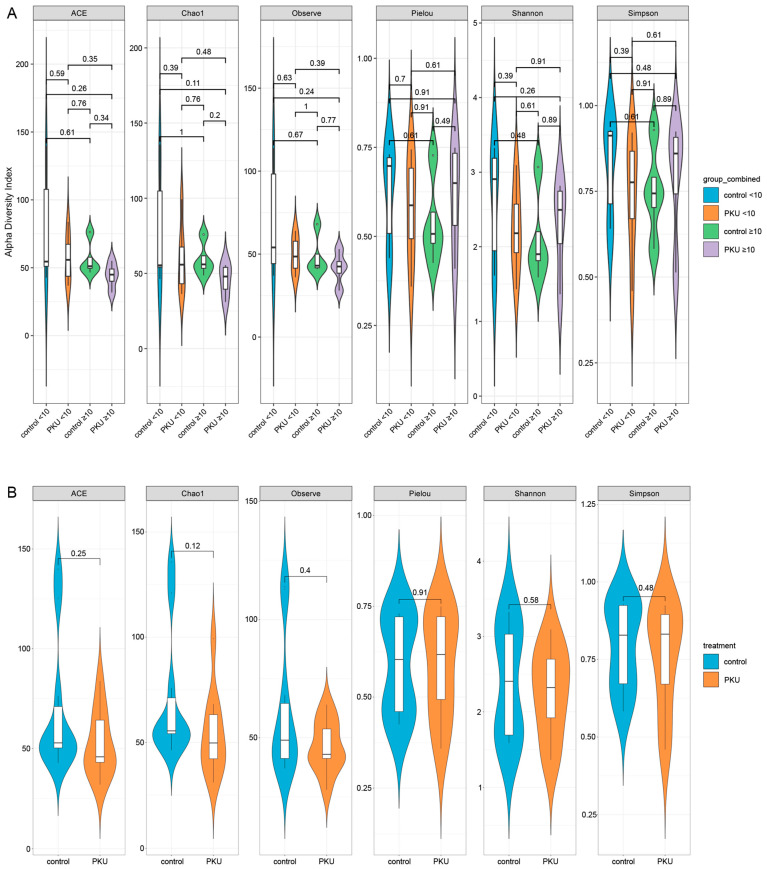
(**A**,**B**) Alpha diversity of the gut mycobiome in PKU and control groups. (**A**,**B**) Violin plots represented the distribution of six alpha diversity indices (ACE, Chao1, Observed, Pielou, Shannon, and Simpson) across four subgroups based on age and disease status: control_ < 10, PKU_ < 10, control_≥10, and PKU_≥10 (**A**). The same diversity indices are presented without age stratification, comparing the overall control and PKU groups (**B**). The white boxplots in each violin plot represented the interquartile range and the median. Statistical analyses among groups utilised the Wilcoxon rank-sum test, with *p*-values indicated above each comparison.

**Figure 2 nutrients-17-02405-f002:**
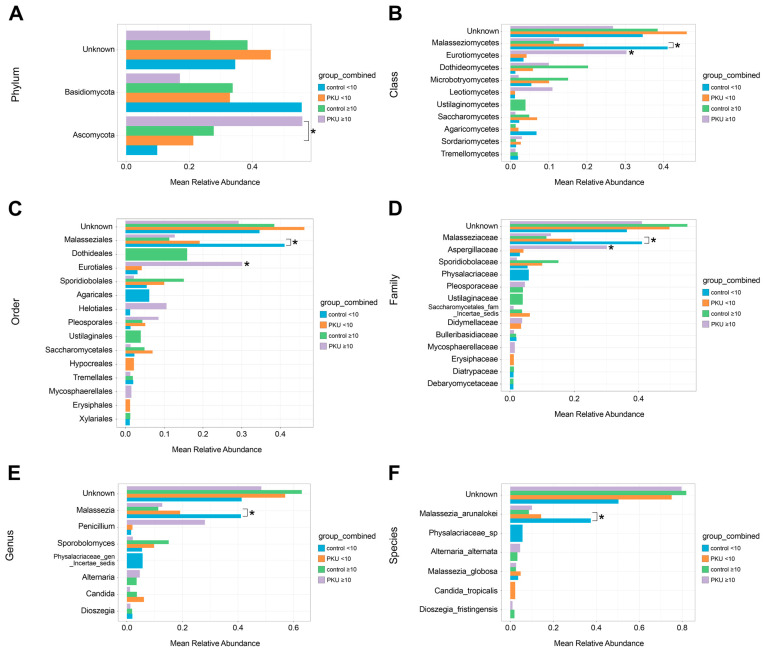
(**A**–**E**) Mean relative abundance of fungi in the gut mycobiome of the PKU group compared to the control group, each split into <10-year and ≥10-year age groups. The bar plots show the mean relative abundance of fungi at each of the six taxonomic levels (>0.01): (**A**) phylum, (**B**) class, (**C**) order, (**D**) family, (**E**) genus, and (**F**) species. The x-axis shows relative abundance, while the y-axis lists the identified fungal taxa. The bars are coloured based on the study group: blue for control_<10, orange for PKU_<10, green for control_≥10, and purple for PKU_≥10. The asterisk indicates a significant difference between groups with a * *p*-value < 0.05.

**Figure 3 nutrients-17-02405-f003:**
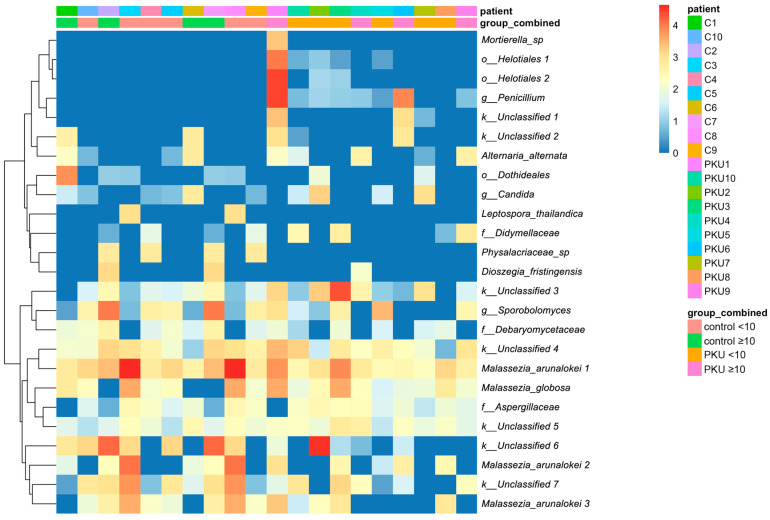
Heatmap of fungal taxon abundance in the control group (C1–C10) compared to the phenylketonuria group (PKU1–PKU10). To show the difference between the 25 most common fungal taxa in the gut mycobiome of healthy children and children with PKU, a logarithmic transformation of 10 was used. The columns represent individual patients, with colour coding based on patient ID and treatment group/age (control_<10 vs. PKU_<10; control_≥10 vs. PKU_≥10). The rows represent fungal taxa, organised hierarchically according to their similarities in abundance. The colour gradient (from red to blue) reflects the relative abundance levels, with red indicating high abundance and blue indicating low or no abundance.

**Figure 4 nutrients-17-02405-f004:**
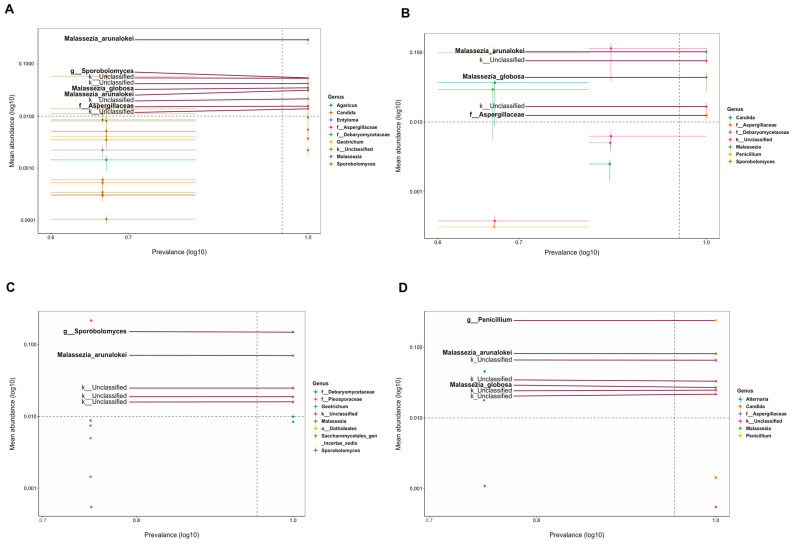
(**A**–**D**) Prevalence and abundance of gut fungal genera in control vs. PKU groups. The scatterplot illustrated the relationship between the prevalence and mean relative abundance of various fungal taxa in the gut mycobiome, with both axes displayed on a logarithmic scale: control_<10 (**A**), PKU_<10 (**B**), control_≥10 (**C**) and PKU_≥10 (**D**). The x-axis shows the prevalence (log10), representing the proportion of samples in which each genus was detected. The y-axis shows the mean relative abundance (log10) of each genus across the samples. The dotted lines in panels A–D represent predefined thresholds for identifying core fungal taxa. Specifically, the vertical line at 0.95 on the x-axis denotes taxa present in at least 95% of samples (high prevalence), and the horizontal line at 0.01 on the y-axis indicates a 1% mean relative abundance threshold. Each point represents a fungal genus; error bars reflect variation within the group. The colours correspond to taxonomic classifications indicated in the legend.

**Figure 5 nutrients-17-02405-f005:**
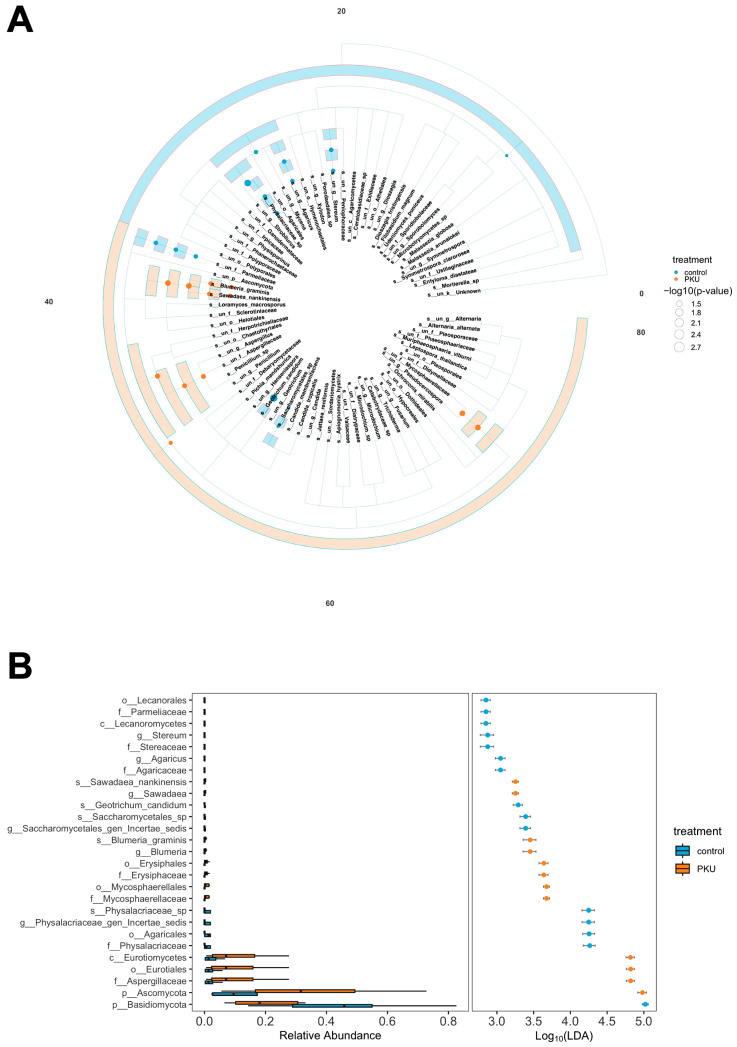
(**A**,**B**) Differential taxonomic abundance analysis between control and PKU groups identified by LEfSe analysis. (**A**) The figure shows a cladogram of fungal taxa with significant differences in abundance between the control (blue) and PKU (orange) groups, identified by linear discriminant analysis (LDA) effect size (LEfSe). The size of the circles indicates the magnitude of the differences, determined by −log10(*p*-value). Larger circles represent lower *p*-values, indicating higher significance. (**B**) Bar plots displaying the relative abundance (left) and the logarithmic LDA scores (log10) (right) for taxa significantly enriched in either control (blue) or PKU (orange) groups. Only taxa meeting the threshold of statistical significance and an LDA score > 2 are shown.

**Figure 6 nutrients-17-02405-f006:**
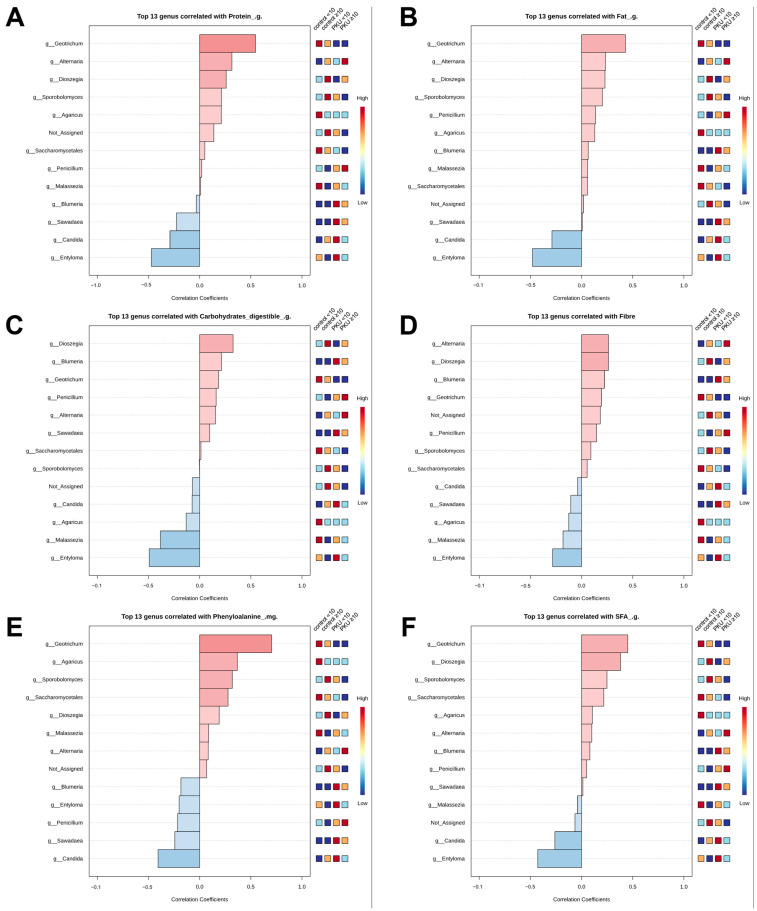
(**A**–**F**) Correlation analysis of gut microbiota and dietary compounds. This figure combines six bar plots showing the top fungal genera most strongly correlated with the intake of different macronutrients: protein (g), fat (g), carbohydrates (digestible, g), fibre (g), saturated fatty acids (SFA, g), and phenylalanine (mg). The plots display the coefficients of Spearman rank correlation, with red bars indicating positive correlations and blue bars negative ones. To the right of each plot, a heatmap depicts relative abundance across subgroups: control_<10, control_≥10, PKU_<10, and PKU_≥10.

**Table 1 nutrients-17-02405-t001:** Demographic, anthropometric, and nutritional characteristics of the participants.

No	Gender	Age	Body Mass (kg)	Height (m)	BMI (kg/m^2^)	Nutritional Status
Control group						
C1	Male	17.0	74.0	1.71	25.3	Overweight
C2	Female	12.8	58.0	1.62	22.1	Norm
C3	Female	8.6	25.0	1.32	14.3	Underweight
C4	Male	3.0	14.0	0.98	14.6	Norm
C5	Female	5.0	22.0	1.16	16.3	Norm
C6	Female	17.0	67.0	1.66	24.3	Norm
C7	Male	13.0	52.0	1.53	22.2	Overweight
C8	Male	6.4	24.0	1.26	15.1	Norm
C9	Male	9.5	38.0	1.42	18.8	Norm
C10	Female	3.4	16.0	1.00	16.0	Norm
AVG	–	9.6 ^A^ ± 5.2	39.0 ^A^ ± 22.1	1.4 ^A^ ± 0.3	18.9 ^A^ ± 4.2	–
Studied group						
PKU1	Male	17.0	70.0	1.70	24.2	Overweight
PKU2	Female	3.0	17.3	1.00	17.3	Norm
PKU3	Female	1.4	12.3	0.86	16.6	Norm
PKU4	Male	13.0	64.0	1.59	25.3	Overweight
PKU5	Male	6.0	25.0	1.24	16.3	Norm
PKU6	Female	17.5	50.6	1.64	19.0	Norm
PKU7	Male	2.9	14.0	0.89	17.7	Norm
PKU8	Male	2.8	11.8	0.86	16.0	Underweight
PKU9	Female	12.7	37.0	1.56	15.2	Norm
PKU10	Female	9.8	33.4	1.36	17.8	Norm
AVG	–	8.6 ^A^ ± 6.2	33.5 ^A^ ± 21.6	1.3 ^A^ ± 0.3	18.5 ^A^ ± 3.5	–

^ABC^ mean scores with different letters in the same column are significantly different at *p* < 0.05 (Student’s *t*-test or Mann–Whitney U test). Abbreviations: AVG—average; BMI—Body Mass Index; Norm—normal weight.

**Table 2 nutrients-17-02405-t002:** Assessment of energy and macronutrient intake in the control group.

No	Energy (kcal)				Protein (g)			Fats (g)		
	Mean Daily Intake	EER	Compliance with the Standard (%)		Mean Daily Intake	10–20% of EER (g)	Compliance with the Standard	Mean Daily Intake	30–40% of EER (g)	Compliance with the Standard
C1	2635.9	2933	89.9		137.5	73.3–146.7	Norm	90.2	97.8–130.4	Norm
C2	2343.8	2111	111.0		82.7	52.8–105.6	Norm	70.2	70.4–93.8	Norm
C3	1854.4	1686	110.0		73.2	42.2–84.3	Norm	58.9	56.2–74.9	Norm
C4	1205.2	1163	103.6		44.2	29.1–58.2	Norm	42.5	38.8–51.7	Norm
C5	1255.8	1419	88.5		41.3	35.5–71.0	Norm	40.4	47.3–63.1	Under
C6	2027.4	2255	89.9		92.5	56.4–112.8	Norm	75.2	75.2–100.2	Norm
C7	1519.8	2384	63.8		65.3	59.6–119.2	Norm	50.3	79.5–106.0	Under
C8	2158.8	1620	133.3		85.2	40.5–81.0	Norm	78.3	54.0–72.0	Norm
C9	1746.6	1934	90.3		65.5	48.4–96.7	Norm	50.0	64.5–86.0	Under
C10	1260.8	1088	115.9		57.0	27.2–54.4	Norm	40.4	36.3–48.4	Norm
AVG	1800.9 ^A^ ± 494.4	–	–		74.4 ^A^ ± 27.8	–	–	59.6 ^A^ ± 17.8	–	–
**No**	**SFA (g)**			**Carbohydrates (g)**			**Dietary Fibre (g)**			**Phe (mg)**
	**Mean Daily Intake**	**6% of EER**	**Compliance with the Standard**	**Mean Daily Intake**	**45–65% of EER (g)**	**Compliance with the Standard**	**Mean Daily Intake**	**AI (g)**	**Compliance with the Standard**	**Mean Daily Intake**
C1	44.1	19.6	Above	326.3	330.0–476.6	Norm	35.9	21	Enough intake	5134.4
C2	28.1	14.1	Above	361.1	237.5–343.0	Norm	35.8	19	Enough intake	3799.6
C3	23.8	11.2	Above	267.3	189.7–274.0	Norm	20.2	16	Enough intake	3061.4
C4	12.1	7.8	Above	165.6	130.8–189.0	Norm	8.3	10	Deficiency	1896.0
C5	17.8	9.5	Above	187.2	159.6–230.6	Norm	14.3	14	Enough intake	1752.8
C6	35.6	15.0	Above	263.1	253.7–366.4	Norm	33.0	21	Enough intake	4048.9
C7	15.4	15.9	Norm	211.7	268.2–387.4	Under	19.2	19	Enough intake	2913.8
C8	41.2	10.8	Above	288.5	182.3–263.3	Norm	19.9	14	Enough intake	3626.7
C9	19.9	12.9	Above	266.2	217.6–314.3	Norm	13.4	16	Deficiency	2992.1
C10	17.1	7.3	Above	175.8	122.4–176.8	Norm	19.9	10	Enough intake	2498.3
AVG	25.5 ^A^ ± 11.3	–	–	251.3 ^A^ ± 65.3	–	–	22.0 ^A^ ± 9.7	–	–	3172.4 ^A^ ± 1023.5

^ABC^ mean scores with different letters in the same column Table 2 and Table 3 are significantly different at *p* < 0.05 (Student’s *t*-test). Abbreviations: EER—Estimated Energy Requirement; AI—adequate intake; Norm—Normal Nutritional Status.

**Table 3 nutrients-17-02405-t003:** Assessment of energy and macronutrient intake in the studied group.

No	Energy (kcal)			Protein (g)				Fats (g)		
	Mean Daily Intake	EER	Compliance with the Standard (%)	Mean Daily Intake	10–20% of EER (g)	Compliance with the Standard		Mean Daily Intake	30–40% of EER (g)	Compliance with the Standard
PKU1	2149.4	2933	136.5	84.2	73.3–146.7	Norm		85.2	97.8–130.4	Under
PKU2	1232.8	1088	88.3	43.5	27.2–54.4	Norm		25.2	36.3–48.4	Under
PKU3	939.4	712	75.8	31.5	17.8–35.6	Norm		46.2	23.7–31.6	Above
PKU4	2752.9	2384	86.6	75.2	59.6–119.2	Norm		90.9	79.5–106.0	Norm
PKU5	1843.6	1620	87.9	50.5	40.5–81.0	Norm		65.4	54.0–72.0	Norm
PKU6	1854.5	2255	121.6	53.2	56.4–112.8	Norm		43.6	75.2–100.2	Under
PKU7	1045.9	1096	104.8	28.0	27.4–54.8	Norm		19.9	36.5–48.7	Under
PKU8	755.3	1096	145.1	18.0	27.4–54.8	Under		25.2	36.5–48.7	Under
PKU9	1654.4	2016	121.9	61.5	50.4–100.8	Norm		41.7	67.2–89.6	Under
PKU10	1205.1	1790	148.5	55.8	44.8–89.5	Norm		28.8	59.7–79.6	Under
AVG	1543.3 ^A^ ± 621.3	–	–	50.1 ^B^ ± 20.8	–	–		47.2 ^A^ ± 25.3	–	–
**No**	**SFA (g)**			**Carbohydrates (g)**			**Dietary Fibre (g)**			**Phe (mg)**
	**Mean Daily Intake**	**6% of EER**	**Compliance with the Standard**	**Mean Daily Intake**	**45–65% of EER (g)**	**Compliance with the Standard**	**Mean Daily Intake**	**AI (g)**	**Compliance with the Standard**	**Mean Daily Intake**
PKU1	27.7	19.6	Above	272.2	330.0–476.6	Under	21.2	21	Enough intake	3211.4
PKU2	8.2	7.3	Above	213.2	122.4–176.8	Above	10.5	10	Enough intake	365.1
PKU3	6.3	4.7	Above	108.3	80.1–115.7	Norm	18.0	10	Enough intake	266.8
PKU4	43.7	15.9	Above	422.2	268.2–387.4	Norm	30.4	19	Enough intake	1150.0
PKU5	25.8	10.8	Above	271.8	182.3–263.3	Norm	17.7	14	Enough intake	777.0
PKU6	20.4	15.0	Above	329.5	253.7–366.4	Norm	35.3	21	Enough intake	471.7
PKU7	5.0	7.3	Norm	194.8	123.3–178.1	Norm	13.5	10	Enough intake	455.1
PKU8	7.8	7.3	Norm	116.7	123.3–178.1	Norm	5.6	10	Deficiency	255.5
PKU9	14.4	13.4	Norm	267.2	226.8–327.6	Norm	18.7	19	Enough intake	439.7
PKU10	10.4	11.9	Norm	192.6	201.4–290.9	Norm	23.8	16	Enough intake	358.9
AVG	17.0 ^A^ ± 12.4	–	–	238.9 ^A^ ± 95.2	–	–	19.5 ^A^ ± 8.9		–	775.1 ^B^ ± 897.7

^ABC^ mean scores with different letters in the same column as Table 2 and Table 3 are significantly different at *p* < 0.05 (Student’s *t*-test). Abbreviations: EER—Estimated Energy Requirement; AI—adequate intake; Norm—Normal Nutritional Status.

## Data Availability

The original ITS sequencing raw data presented in the study are openly available in NCBI SRA under accession number PRJNA1257280. The data are available at: https://www.ncbi.nlm.nih.gov/bioproject/PRJNA1257280 (accessed on 21 July 2025).

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
