# Peer review of "The Gut Mycobiome and Nutritional Status in Paediatric Phenylketonuria: A Cross-Sectional Pilot Study"

_nutrients, 2025, doi:10.3390/nu17152405_

Round 1

Reviewer 1 Report

Comments and Suggestions for Authors

Summary:

This preliminary study explores the gut mycobiome in children with phenylketonuria (PKU), an underexplored area in microbiome research. By comparing the fungal profiles of PKU patients with those of age-matched healthy controls, the study provides preliminary results suggesting potential interactions between dietary management, fungal composition, and metabolic health. The manuscript is well-written and contributes meaningfully to the emerging understanding of host–fungi interactions in rare metabolic disorders. I congratulate the authors for addressing such a novel and relevant topic. However, I believe some areas could be strengthened to enhance the manuscript’s impact.

Comments:

Introduction section

  1. Please check the grammar on line 65 and review the entire sentence (lines 63–66), as it appears redundant.
  2. A period is missing at the end of the sentence that concludes on line 119.

Results section

The results are well described and hold significant scientific relevance. I recommend revisiting the text to reduce redundancies.

Discussion section

  1. I believe subsection 4.1 would benefit from a deeper discussion on how dietary patterns may influence the nutritional status of children with PKU, for example, by referencing findings from other studies.

Reviewer 2 Report

Comments and Suggestions for Authors

I have read this paper with interest, and with a background on perinatal and neonatal clinical research. I perceive this pilot study as rather hypothesis generating, and of potential value to the PKU research and care. The specific focus on fungal (mycobiome) does provide additional novelty. I hereby value the use of association, not necessary causal.

As I’m aware that (partial) breastfeeding has been implemented as part of PKU care, do you have data on previous breastfeeding in included cases and controls (since BF may affect microbiome, or was this not yet the cases when these PKU cases and controls were born?

Can/dare the authors to speculate on the mechanisms related to the differences observed, e.g. ‘recent’ antibiotics, low protein diet with low phenylalanine ?

Have the authors considered to assess body composition besides the BMI measurement ?

How have you decided on the 3 day food diary. Was this rather based on feasibility, or is there evidence that 3 days is ‘sufficient’ to associated with microbiome, assuming that there is less fluctuations in microbiome versus daily intake ?

The bioanalytical part reads accurate in my assessment, be it that I do not perceive me to be an expert on this specific methodology.

Being aware of the small sample size, but how confident are you that the cases and controls are representing reasonable well their population(s) ?

Editing

Figure 1, legend, suggest to add the age subgroup information for transparency. Similar comment for legend figure 6, and please use (throughout paper and figures) consistent >10 or ≥10 (is at present 10 and 11 year is ‘absent’, is either < 10 or >11)(suggest to also check the supplemental materials on this)

Reviewer 3 Report

Comments and Suggestions for Authors

I believe the manuscript submitted to Nutrients by Ostrowska et al. can be considered for publication after some revisions. These are my suggestions:

The abstract is too long. Please make it more concise, according to the journal’s guidelines. Highlight your main results in a quantitative way and indicate future perspectives and some directions for further studies.

Abbreviations should be avoided as keywords.

Lines 51-53: References are missing.

Some of your references are not formatted properly. See the example in line 92.

The relevance of this study from an international perspective has to be clarified in the Introduction.

The stool sample collection and preservation need to be detailed.

It would be useful to provide a flowchart with the steps taken to conduct all the experiments.

Figure 4 should be improved. It is hard to understand.

The Results and Discussion sections are well described. However, I recommend a further analysis of your study’s limitations.

The Conclusions can also be improved, taking into consideration an international overview and the practical implications of the current research.

Author Response

Comment 1: I believe the manuscript submitted to Nutrients by Ostrowska et al. can be considered for publication after some revisions. These are my suggestions:

The abstract is too long. Please make it more concise, according to the journal’s guidelines. Highlight your main results in a quantitative way and indicate future perspectives and some directions for further studies.

Response 1: We thank the Reviewer for the valuable suggestion. In response, we have revised the abstract to ensure conciseness and compliance with the journal’s word limit. The key findings are now presented in a more quantitative manner, emphasising statistically relevant results (e.g., Eurotiales ↑ in PKU, p = 0.029; Geotrichum–phenylalanine correlation ρ = 0.70, p = 0.0005). We also added a final sentence to show how fungal markers could be useful in medicine and to recommend future research on tracking and studying the gut mycobiome in PKU over time. The revised abstract appears on page 1, lines 24-42, of the manuscript.

Comment 2: Abbreviations should be avoided as keywords.

Response 2: We thank the Reviewer for the comment regarding the use of abbreviations in the keywords section. In response, we have revised the keywords to avoid standalone abbreviations, replacing them with their full forms, as follows (page 1, lines 43-44):

“phenylketonuria (PKU); gut mycobiome; internal transcribed spacer (ITS) sequencing; phenylketonuria diet; paediatric nutrition; fungal dysbiosis”

Comment 3: Lines 51-53: References are missing.

Response 3: We thank the Reviewer for this suggestion. Citations have been added ([1, 2] – line 49; page 2):

  1. Hoffmann, C.; Dollive, S.; Grunberg, S.; Chen, J.; Li, H.; Wu, G.D.; Lewis, J.D.; Bushman, F.D. Archaea and Fungi of the Human Gut Microbiome: Correlations with Diet and Bacterial Residents. PLoS One 2013, 8, e66019, doi:10.1371/JOURNAL.PONE.0066019.
  1. Seed, P.C. The Human Mycobiome. Cold Spring Harb Perspect Med 2015, 5, a019810, doi:10.1101/CSHPERSPECT.A019810.

Comment 4: Some of your references are not formatted properly. See the example in line 92.

Response 4: We thank the Reviewer for pointing out the issue with reference formatting. We have carefully reviewed all citations and adjusted them to conform to the journal’s referencing style. Specifically, the formatting has been corrected to match the example provided (page 2, line 92), and we ensured consistency across the entire reference list and in-text citations.

The citation format has also been corrected in other parts of the publication (page 3, line 126).

Comment 5: The relevance of this study from an international perspective has to be clarified in the Introduction.

Response 5:  We thank the Reviewer for this important observation. In response, we have revised the introduction to better highlight the international relevance of our study. Specifically, we added the following sentences:

Page 2, lines 65-67: “This global variability underscores the relevance of investigating PKU from an international perspective, particularly when considering the shared dietary strategies used worldwide.”

Page 3, lines 112-121: “While international dietary guidelines for PKU are well established, there is increasing recognition that dietary regimens may have broader systemic effects, particularly in the context of gut microbial ecosystems. Although most research has focused on bacterial microbiota, recent studies highlight the potential role of the gut mycobiome—the fungal component of the microbiota—in modulating host immunity, metabolism, and inflammatory status. However, data on the gut mycobiota in PKU populations remain extremely limited, especially in children. By examining both fungal composition and nutritional intake in a paediatric PKU cohort, our study aims to contribute new insights that may be of value to the global discussion on personalised diet–microbiome interactions in metabolic disorders.”

These additions aim to clarify how our findings may inform international efforts to understand the systemic effects of dietary management at PKU and support the growing field of personalised microbiome-based interventions.

Comment 6: The stool sample collection and preservation need to be detailed.

Response 6: We thank the Reviewer for the suggestion. We have revised the section to include additional details on stool sample collection and preservation (page 4, lines 157-165)

“Stool samples were collected at the time of dietary data acquisition using the Stool DNA Collection & Stabilization Kit (Canvax, Valladolid, Spain). This all-in-one system enables fast, hygienic self-collection and contains 8 mL of a proprietary stabilisation buffer that inactivates nucleases and prevents microbial overgrowth, preserving native microbial community structure. Participants and their legal guardians received written and verbal instructions regarding hygienic self-collection procedures. The integrated sampling spoon and wide-mouth tube design ensured user-friendly, contamination-minimised collection. Samples were stored at room temperature (15-25 °C) for up to 48 hours before transfer to -20 °C, where they remained until further analysis.”

Comment 7: It would be useful to provide a flowchart with the steps taken to conduct all the experiments.

Response 7: We thank the Reviewer for this thoughtful suggestion. While we believe that the methodology is described in sufficient detail in the Methods section and follows a clear logical structure, we agree that a visual summary can be advantageous for review purposes. Therefore, although we do not plan to include a flowchart in the final version of the manuscript, we have prepared a schematic diagram outlining the experimental workflow to assist in the review process (attached separately).

Comment 8: Figure 4 should be improved. It is hard to understand.

Response 8: We thank the Reviewer for the constructive feedback regarding Figure 4. In response, we have revised the figure and updated its legend to enhance clarity and interpretability. Specifically, we improved axis labelling, increased the legibility of genus names, and added a more detailed and descriptive caption (page 15, lines 454-464).

“Figure 4. (A-D) Prevalence and abundance of gut fungal genera in control vs. PKU groups. The scatterplot illustrated the relationship between the prevalence and mean relative abundance of various fungal taxa in the gut mycobiome, with both axes displayed on a logarithmic scale: control_<10 (A), PKU_<10 (B), control_≥10 (C) and PKU_≥10 (D). The x-axis shows the prevalence (log10), representing the proportion of samples in which each genus was detected. The y-axis shows the mean relative abundance (log10) of each genus across the samples. The dotted lines in panels A–D represent predefined thresholds for identifying core fungal taxa. Specifically, the vertical line at 0.95 on the x-axis denotes taxa present in at least 95% of samples (high prevalence), and the horizontal line at 0.01 on the y-axis indicates a 1% mean relative abundance threshold. Each point represents a fungal genus; error bars reflect variation within the group. The colours correspond to taxonomic classifications indicated in the legend.”

All instances of p-values have been formatted in italics (e.g., p = 0.013), in accordance with journal style. The age grouping previously indicated as ">11" has been corrected to "≥10" throughout the manuscript for clarity and consistency. All changes in the main text are highlighted in blue, while updates in the supplementary materials are marked in blue font for easy identification

Comment 9: The Results and Discussion sections are well described. However, I recommend a further analysis of your study’s limitations.

Response 9: We thank the Reviewer for highlighting the importance of providing a more comprehensive overview of the study’s limitations. In response, we have expanded the Discussion section to include a dedicated paragraph entitled “Study Limitations” (see page 25, lines 790–811). This subsection now addresses several key aspects: the limitations of our cross-sectional design; the modest sample size after subgroup stratification; the potential biases inherent in short-term dietary recall; and the limitations of ITS sequencing for fungal profiling, including low fungal biomass and incomplete taxonomic resolution. We have also acknowledged that although the study was conducted in a single national population, generalisability may be limited. Importantly, in light of potential microbial confounders, we clarify that all participants were screened for recent antimicrobial or probiotic use, and individuals who had taken antibiotics or probiotics within the three months preceding sample collection were excluded from the study. We believe this additional discussion enhances the transparency and scientific rigour of the manuscript.

Comment 10: The Conclusions can also be improved, taking into consideration an international overview and the practical implications of the current research.

Response 10: We thank the Reviewer for this valuable comment. As suggested, the Conclusions section has been revised to include an international perspective and to highlight the practical implications of our findings (page 26, lines 852-866).

  1. Response to Comments on the Quality of English Language

Not applicable

  1. Additional clarifications

Any changes made directly to the manuscript have been highlighted in blue.
